# COVID-19 guidelines and its perceived effect on seafarers' health and wellbeing: A qualitative study

**Amit Timilsina**[1,2] *, **Fereshteh Baygi**[3]

**1** Central Department of Public Health, Tribhuvan University, Kathmandu, Nepal, **2** Department of Public Health, University of Southern Denmark, Esbjerg, Denmark, **3** Department of Public Health, Research Unit of General Practice, University of Southern Denmark, Odense, Denmark

☯ These authors contributed equally to this work.
* timilsinaamit@gmail.com

## Abstract

### Background

The COVID-19 pandemic and its guidelines have had a profound impact on the social life, health, and wellbeing of people around the world. Very little is known if the guidelines are put into action effectively by seafarers. Also, the effects of such guidelines on seafarers' health and wellbeing have not been studied so far. Therefore, this study aimed to explore the perceived effects of the COVID-19 guidelines on seafarers' health and wellbeing.

### Materials & methods

A qualitative research method was adopted using in-depth interviews. A total of 13 international male seafarers were interviewed until data saturation was achieved. Purposive sampling was used to recruit the respondents. The data was inductively coded using NVivo 12 and manifest content analysis was conducted.

### Results

Once seafarers had access to COVID-19 guidelines, seafarers followed the guideline as suggested by companies under the guidance of captain. The majority of the participants reported that available guidelines could decrease their stress and anxiety levels, while some reported that implementing the COVID-19 guidelines had no significant effect on their health and wellbeing. The results of this study also showed a delay in repatriation, which had an impact on the seafarers' mental health.

### Conclusion

The guidelines could not address seafarers' psychological needs to ensure their good health and wellbeing. Therefore, it is recommended that shipping companies address the mental

**Data Availability Statement:** All relevant data are within the paper.

**Funding:** The authors received no specific funding for this work.

**Competing interests:** The authors have declared that no competing interests exist.

**Abbreviations: BCC**, Behaviour Changing Communications; **COVID**, Coronavirus Disease of 2019; **DPA**, Designated Person Ashore; **EMSA**, European Maritime Safety Agency; **ERS**, Emergency Response System; **EU**, European Union; **FLP**, Freight, Logistics and Passenger; **GDPR**, General Data Protection Regulation; **ICS**, International Chamber of Shipping; **IHR**, International Health Regulation; **IMO**, International Maritime Organization; **MLC**, Maritime Labor Convention; **PCR**, Polymerase Chain Reaction; **PPE**, Personal Protective Equipment; **SOLAS**, Safety of Life at Sea; **STCW**, Standards of Training, Certification and Watch keeping for Seafarers; **TMAS**, Telemedical Maritime Assistance Service; **USA**, United States of America; **WHO**, World Health Organization.

health needs of seafarers during the COVID-19 pandemic and similar infectious diseases emerging in the future.

## 1. Introduction

Coronavirus disease 2019 (COVID-19) has an impact on society, health, economy, and the environment [1]. The social relationship have been stranded, health status has deteriorated,, the health system has been strained and people's lives has been challenged in many ways [1].

The maritime sector, like any other sector also has been affected by the COVID-19 pandemic [2]. After the detection of the first case of COVID-19 on 31st December 2019, the World Health Organization (WHO) declared COVID-19 as a public health emergency on 30th January 2020 [3]. The first case of COVID-19 was observed in the maritime sector, where one passenger was found to be positive for the virus on the cruise ship Diamond Princess [4]. Many countries around the world went into lockdown and turned back many ships coming from shore [5]. Seafarers were also stranded on board for 5–6 months on a ship that had the biggest toll on the seafarers' mental health [6]. Due to a failure in crew change, seafarers were bound to work for a long time at sea beyond the working limits as set by Maritime Labor Convention (MLC) 2006. The ship-owner also faced inadequate support required to facilitate the process [7].

During the COVID-19 pandemic, excessive work-related stress, fear due to insecurities, and a considerable sense of concern for the family and management issues have had a significant impact on the seafarers' health and wellbeing [8]. Seafarers' health and wellbeing are important because they contribute to the attenuation of quality of work, leading to an increased risk of accidents, injuries, suicides, and high turnover of staff due to dissatisfaction at work [9]. In the support of international maritime organizations WHO provides guidance on caring for patients infected with COVID-19 to minimize the risk of COVID-19 [10]. Furthermore, the International Chamber of Shipping (ICS) issues COVID-19 Guidance for ship operators for the Protection of the Health of Seafarers in association with the International Maritime Organization (IMO) and WHO [11]. Implementation of the COVID-19 preventive measures and guidelines in a stressful work environment on board the vessels might affect the health and wellbeing of the seafarers. Although guidelines are in place, we know very little about the adaptability and implementation of the guidelines. Thus, this study aimed to understand the implementation of the health guidelines in the prevention of COVID-19 among seafarers and shipping companies and to explore their effects on the seafarers' health and wellbeing.

## 2. Materials and methods

### 2.1. Study setting and design

This study was conducted on managers of two international shipping companies and seafarers of long-voyage vessels. The working departments on such vessels include deck, engine, and service departments. The crew consists of officers, technicians, and rankers.

This qualitative study was conducted using content analysis to explore the effects of the implementation of the COVID-19 guidelines on the participants' health and wellbeing. In-depth interviews were conducted to collect primary data using a semi-structured interview guide. Content Analysis was used to analyse the primary data, which was transcribed [12].

## 2.2. Data collection and participants

Similarly, a detailed literature review of available guidance and studies conducted in the maritime sector during the COVID-19 period was conducted. For this, five available guidelines were reviewed, the important information were extracted in a matrix on an excel sheet, and it was analysed to understand the background and to develop tools.

A total of ten seafarers who worked on ocean-going vessels and held different ranks on ships and three managers of two international shipping companies were interviewed. This study was approved by University of Southern Denmark (No. 11.274). A purposive sampling design was used to select participants, in which participants were contacted, the purpose of the study was explained, and a consent form was sent to participants, which they accepted after receiving it. Thus, written consent was obtained prior to the interview using a written consent form. Besides this, verbal informed consent was also obtained from all participants before starting the interview and the objectives of the research were re-explained to them.

The data was collected in February 2021 for a month. Seafarers who were on board during the COVID-19 pandemic were recruited for study purposes. Participants who were comfortable to give interviews in English were recruited in this study, so all interviews were conducted in the English language. An in-depth telephonic interview was conducted to address the research questions. All interviews were conducted online using WhatsApp and were recorded using a laptop.

The semi-structured interview guide was developed before the data collection. The data was collected using semi-structured interviews. Participants were asked socio-demographic questions, including questions related to the implementation of COVID-19 guidelines and experiences of seafarers regarding the effects of COVID-19 preventive measures and guidelines on their health and wellbeing. The experiences of managers are also important, particularly to understand the context. Follow-up questions were asked with some premeditated and ad hoc questions based on the participants' answers and experiences.

The average length of an interview was 40 minutes. The data was collected until the data saturation was achieved. Data saturation for this study was reached as no new information was attained and there was enough data to replicate the research study [12]. One of the methods used to ensure data saturation is to through data triangulation. The data triangulation was conducted with multiple data sources, including three managers who worked at various shipping companies and were interviewed after the completion of seven in-depth interviews. Some new information was obtained from the data triangulation, and three more in-depth interviews were conducted. The findings of the literature review and the interview with the managers were in line with the findings of the in-depth interview with the seafarers. The findings from In-depth Interview (IDI) resonated with the findings of the analysis of guidance leading to redundancy of information and thus suggesting data saturation.

## 2.3. Coding and analysis

Qualitative content analysis has been used as an analysis method to interpret the meaning of primary data derived from the content of the interview [13]. All the recordings were transcribed word by word and saved with appropriate pseudonymized file names in a secure place. The NVivo 12 was used to organize and analyse the transcribed text data of the interview [14]. The inductive analysis process has been used to organize the textual data, including open coding, creating a coding sheet, grouping and categorization, and abstraction of the data [15].

The original text was read and re-read, and the interview text was condensed into two content areas: 1) Implementation of health guidelines in the prevention of COVID-19 and 2) Effects of COVID-19 preventive measures and guidelines on the seafarers' health and wellbeing.

Similarly, the files were read and re-read, and initial points were noted before developing the codes. The text interview was divided into meaning units and condensed. The condensed meaning was abstracted and labelled with a code. The codes were created from the text in NVivo in relation to the context of the data (also known as the open coding process). The generated codes were compared with each other based on their similarities and differences and categorized into themes and subthemes. The code-sharing commonalities were grouped into subthemes, and similar sub-themes were arranged to develop themes. Thus, the themes and subthemes presented in this study were externally heterogeneous and internally homogeneous. The open codebook was discussed together with the supervisor and necessary changes were made.

### 2.4. Trustworthiness of data

The correct operational measure, i.e., content analysis, selection of participants, data collection approach, the number of interviews and the analysis process has been credible as the steps for qualitative content analysis as described by Graneheim and Lund [13, 16]. The semi-structured interview guide was prepared and discussed by the supervisor. The average length of the interview was 40 minutes. The triangulation of data was done by interviewing three managers and reviewing the guidelines. Similarly, the produced code was revised by the supervisor and the theme and sub-theme were decided. These processes have strengthened the credibility to ensure its internal validity [16]. During the in-depth interview, field notes were taken to note important information and process logs. De-briefing for each interview was done by the supervisor.

### 2.5. Ethical consideration

The data was collected and processed in accordance with the General Data Protection Regulation (GDPR) rules. The research follows Helsinki guidelines to ensure optimal ethics before, during and after the research period [17]. Participants were informed about their participation and also written as well as verbal informed consent was obtained from all participants.

## 3. Results

### 3.1. Findings from analysis of guidance

A total of five guidelines available during the study period were reviewed and analysed. The COVID-19 guidance for ship operators for the protection of health and seafarers by the International Chamber of Shipping was issued on March 2020. The guidance includes safety measures before embarkation, on board, after embarkation and explains briefly the mental health of seafarer due to COVID-19 and emphasises using the ISWAN mental health tools available for seafarers [11].

The WHO interim guideline, issued on August 25th, 2020, and the industry recommended framework of protocols for ensuring safe ship changes and travel during COVID-19, issued on December 2nd, 2020, explains step-by-step before embarkation, on board, and after embarkation, respectively. The WHO interim guidance also explains communication procedures and mental health and psychosocial support in brief, while the industry's recommended framework of protocols for ensuring safe ship changes and travel during COVID-19, issued on December 2nd, 2020 [10, 19]. Similarly, the COVID-19 contingency plan and guidelines for the services provided to seafarers and ship owners were issued on 20th March 2020 by the Ministry of Infrastructure and Water Management of the Netherlands [18]. It includes sections such as ship certificates and surveys, musterlists and drills, exemption to the minimum

safe manning document, revalidation of certificates of competency and proficiency, seagoing services, medical certificates, medical equipment, ship-owner's liability and port state control. Also, Interim Guidance for Ships on Managing Suspected or confirmed cases of Coranavirus Disease 2019 was issued on 13[th] February 2020 by the Centre for Disease Control and Prevention, United States of America (USA) [19]. It provides guidance on plans to mitigate COVID-19 on board ships, preventive measures for ship operators, preventive measures for people on board, preboarding procedures, testing for COVID-19, isolation and quarantine procedures and managing non-cruise ships with confirmed cases of COVID-19 and supplies [19].

The findings of these guidelines are consistent across all guidance, including procedures such as masks, gloves, goggles, visors, and Personal Protective Equipment (PPE), social distancing, quarantine and isolation, use of outer walkways by shore personnel, pre-boarding screening, accommodation, and food hygiene practice, and managing COVID-19 suspects and shore side medical services, including Telemedical Maritime Assistance Service (TMAS). WHO interim guidance and COVID-19 guidance for ship operators for the protection of the health of seafarers discuss the mental health of seafarers but fail to provide specific guidance regarding the mental health. Similarly, interim guidance for ships on managing suspected or confirmed COVID-19 cases asserts medical care components to be on board and thus only recommends emergency medical services at port. The COVID-19 contingency plan and guidelines for the services provided to seafarers and ship-owners state that the ship-owners are liable to cover any expenses related to repatriation and ask seafarers to contact the Human Environment and Transport Inspectorate for assistance. However, other available guidance does not emphasize the repatriation process.

## 3.2. Findings from qualitative data

Data saturation was reached after conducting interviews with 13 male seafarers. The age range of the participants was 29–42 years, and the sample of seafarers presented a range of seafaring experience from 8 to 20 years. Similarly, most respondents were married and had children. Of the 13 participants, eight were from the deck, two were from the engine, and three were managers.

A total of forty-three codes were developed from interview texts to answer the two research questions. The codes were further compared and categorized into a theme, which is the implementation of the COVID-19 guideline. There are two categories and three sub-categories within the theme. Management of the guidelines and Effectiveness of the guidelines are two subcategories. Fig 1 shows the relationship between various sub-themes and themes.

**3.2.1. Management of the guideline.** In this section, we discuss the development and revision process of the COVID-19 guidelines. Similarly, this paper presents the availability of guidelines as well as their adequacy for the participant. This section also presents the communication and monitoring mechanisms and process of the implementation of the COVID-19 guideline.

*3.2.1.1. Development and revision.* The majority of the seafarers who used a standard guideline used IMO or WHO or both as reference guidelines. Seafarers also reported using multiple guidelines based on local law, vessel type, and company rules. Seafarers who had access to the guidelines followed them meticulously. R9 says:

"*There are different guidelines. We are starting to follow the International Maritime Organization guidelines which was introduced and published at the beginning of the COVID-19 pandemic.*".

*(R9)*

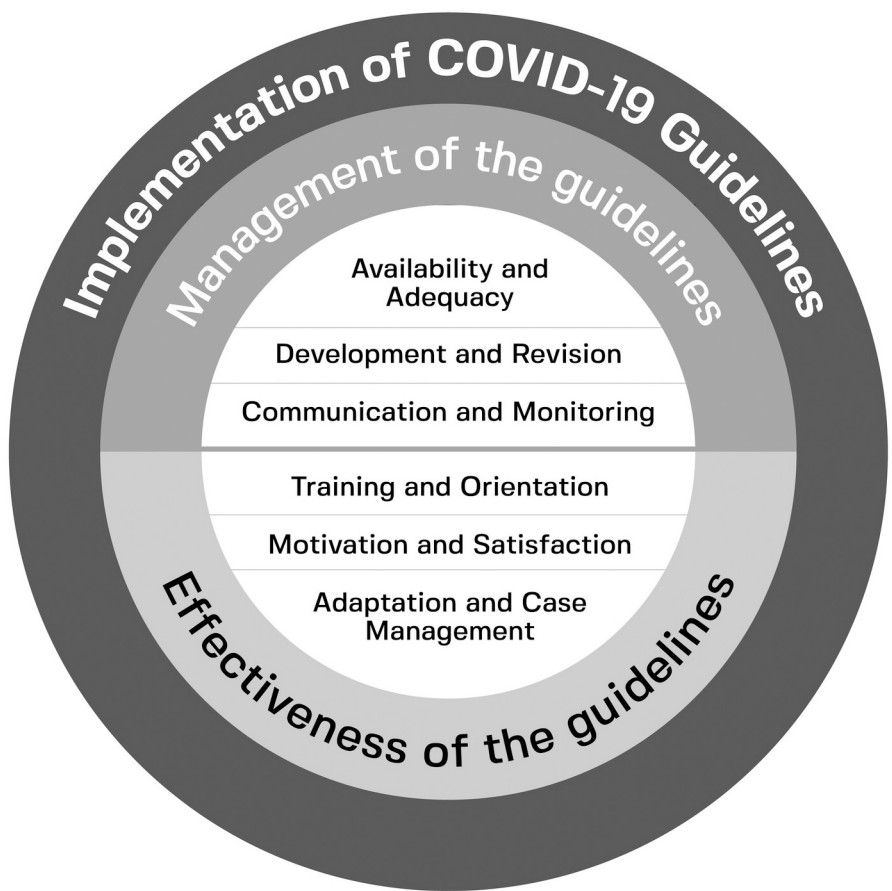

**Fig 1. Pictorial illustration of codebook stating theme and its subtheme.**

There was no management plan in place before the COVID-19 pandemic, although the company has previously dealt with SARS and other infections. The maritime company provided a consolidated guideline in the prevention of COVID-19 based on available standard guidelines, local guidelines, and other available information and circulated it to protect seafarers from COVID-19 infection and for better health and wellbeing. The master of a ship prepared guidelines for their crew at the beginning of the COVID-19 pandemic and the maritime companies also provided consolidated guidelines for the prevention of COVID-19. An example was given by R1:

*"The information was coming from companies which were abstracted from so much research, information from the internet and they were doing some conclusion. It was the only thing available at the time so we cannot expect more at that time. That was enough.".*

*(R1)*

Some participants reported that the guidelines in the prevention of COVID-19 were living documents but not concrete ones. The guideline and regulations were contextually revised to meet the needs of seafarers, local authorities, and trading areas. For instance, R9 says:

*"But case by case we provide them and develop guidelines in the prevention of COVID-19 for different trading areas. At first the guideline was okay just to let them know and take their attention and belief. But we improved and once different cases were coming up, we developed the guideline."*.

*(R9)*

*3.2.1.2. Availability and adequacy.* Participants reported that the guidelines were available after six months and those who joined the ship later had access to standard guidelines. However, the majority of the seafarers who were on board the ship at the start of the COVID-19 pandemic did not receive standard guidelines. The master of the ship made their own guidelines at the beginning of the COVID-19 pandemic based on available information on the internet and their own consciousness, In this regard R13 says:

*"Actually, for guidelines, first of all it was the beginning of the COVID-19 pandemic all around the world and all the companies were facing the problem regarding repatriation of Seafarers and dealing with the masters. There was no strict guideline at the beginning but afterward there was a guideline for the training of the people."*.

*(R13)*

There were mixed responses regarding the adequacy of the information. The majority of the participants believed that the information provided on guidelines was adequate, and the rest of the respondents reported that it was good but not adequate as shared by R2.

*"Information provided by the company was very good but not enough. We searched the social media and read some articles and some more information about that. "*.

*(R2)*

The majority of the participants relied heavily on social media and the internet for their information. Some of the participants were concerned about the authenticity of the news on social media and the internet. Thus, they had confidence in guidelines shared by the company as R3 also shares:

*"We are not following the social media because in social media we are not able to understand whether it is fake or good news so better we must follow the company instructions because the company is not giving the false information. We can believe in company and IMO rather than social media."*.

*(R3)*

Seafarers who have had limited access to information provided by the company or seafarers with less trust in guidelines were found to rely on social media for information regarding COVID-19 and safety measures. R10 shares his experience:

*"Nowadays with this internet access and social media, everybody whatever we are receiving we already knew through our contact or own within."*.

*(R10)*

*3.2.1.3. Communication and monitoring.* In this study, we found that the implementation of COVID-19 guidelines was often monitored by the master. The master is also the communication link between the company and the ship. However, other members were also involved in the monitoring of the process. The master reported the body temperature of the crew, any medical advice required, any equipment required for the seafarers. The companies also had a Freight, Logistics and Passenger (FLP) department which was a bridge between the master and the head of the company. As an example, R13 shares:

*"The master reports regarding the personnel and we trust them. The master is representative of our company onboard the vessel. So, what we did, we collect feedback from master if or not guideline or instruction is followed or not. Because if something happens, unfortunately it also happened on one of our local ports, the master is directly responsible for the matter. ".*

*(R13)*

Seafarers communicated at various levels and with various people during their journey starting from an agent, their crew, office management, shore person and medical care provider. The companies communicated with the seafarers through designated personnel to orient them regarding the guidelines and safety measures as R5 states:

*"Basically, our company has DPA (Designated Person Ashore). So, this DPA's responsibility is to circulate these types of messages to all the fleet. Basically, my company is having a fleet of 31 vessel including tankers and other types of the vessels. So, this DPA is responsible, and he was doing all these things for us. ".*

*(R5)*

Furthermore, on board, the master and other high-ranking officers communicated clear instructions and provided information about the COVID-19 pandemic, as well as reminded everyone of the guidelines to follow and updated any build-up information. Seafarers were also provided a helpline in case they had any emergencies by the companies. There was trust among the seafarers towards the master and they reported any change in temperature, symptoms or support required. For instance, masters were found to be welcoming and considerate of the junior staff, R1reports:

*"Actually, on-board the ship we had people who used to come even with 1 degree rise in temperature and we were also thankful to them so that the next people can come to us easily. We did not have 1–2 cases and also took distance advice to seafarers and also from company. ".*

*(R1)*

Seafarers also received information regarding COVID-19 through social media and the internet. Seafarers communicated with their friends and family members through social media. They communicated with their friends and reached out to personal doctors for the information, like one of the respondents (R3):

*"Then I have to call one of the guys working in Hong Kong, they told me not to worry and its only for TV news basically the ground reporting is something else. I asked him it is safe to join*

*in Hongkong he told me okay it is very safe no problem but take your precaution as much as you can. Then I mentally prepared and I join in Hong Kong".*

(R3)

**3.2.2. Effectiveness of guideline.** This section presents perceived effectiveness among participants of this research based on training and orientation, which provided motivation and satisfaction among seafarers regarding the guidelines and the adaptation and case management.

*3.2.2.1. Training and orientation.* The majority of respondents said they received their training and orientation online via emails, CDs, and videos, as well as from the ship's Master. One of the companies also arranged physical drills regarding safety measures upon request of the master with full precaution on board. Thus, it was evident that the demand and modality of training were related to the needs from seafarers and the leadership of companies. R8 stated that:

*"Company during COVID19 pandemic send the email, through email or circular for this COVID 19 which is read by ship staff and explain by management team to the junior seafarers.".*

(R8)

*3.2.2.2. Motivation and satisfaction.* Resilience was found among the majority of the seafarers accepting the situation and looking forward to a positive outcome, while some of the seafarers were hopeless about the situation and unmotivated. The high-ranking officers also led by example by complying with the guidelines and trying to remain calm during the COVID-19 pandemic and helping others overcome it.

R2 regarding his motivation during the COVID-19 pandemic period at sea says:

*"Most of things which I always do with my crew I always give them some motivation such as you have job at the moment maybe it is possible to be more than you have COVID19, and you will be home and you should be quarantined, and you have problem with family and also you cannot join to ship you will lose your job and many things".*

(R2)

The company offered support and helped to motivate seafarers during the COVID-19 pandemic. They provided them with recreational items so that seafarers could entertain themselves during the COVID-19 pandemic. The master also arranged some on-board programs to cheer up the seafarers. Seafarers were motivated to go to the gym and talk with their families as well.

*"We try to manage them and give them some bonuses to if we could try to sign off them in different ports."*

(R11)

The company also provided other recreational items, such as gym items, game equipment, CDs and DVDs and any other materials as required by the seafarers.

*"We provided them some gym items like weights, like trade mills, we provide them with cards, even PS4, PS5. We provide this kind of item and let them enjoy at least a bit. So just take their mind off the situation, we have sport, as I told you we gave them bonus, we provide them with some gym items, they could have even football or basketball or different items to just make shoot them somehow. ".*

*(R11)*

Seafarers who were stranded at sea were supported with bonuses and internet facilities. The incentives were a big motivation for seafarers during the COVID-19 pandemic to console them. However, the incentive practice was not uniform across the respondents. Some of them received the bonus, while some did not.

*"To the crew, who could not sign off, we gave them some internet bonus and we add salary bonus by 15% to make them little bit happy. We increased the traffic of the internet the free traffic. Because if they use for some downloading and other thing, they have to pay for that but for other communication services, for example, WhatsApp, telegram, IMO or other things we make them free. ".*

*(R12)*

The company supported the family during such a crisis. Seafarers who were infected with COVID-19 were offered monetary incentives and medical support from companies. Furthermore, seafarers also helped their colleagues who could not join the vessel due to the COVID-19 pandemic as their fellow counterparts went through a financial crisis.

*"I have helped out my colleagues. I am not admiring myself but there was situation where my colleagues' bank account goes negative. They are saying that as there is no hope of joining how they can feed their family. Most of the people try to help the seafarers like I helped the other person. ".*

*(R5)*

Seafarers and managers reported that repatriation was the major problem due to the COVID-19 pandemic, where seafarers were stranded for a long time at sea. Seafarers had mental issues due to repatriation and were worried about their families. Both the seafarers and their family members were also worried about them. Seafarers were waiting to be repatriated and go back to their homes. However, neither the seafarers nor the company had any answer regarding when they could leave the sea. Seafaring is a very stressful job, and the issue of repatriation has pushed seafaring further. These uncertainties create frustration and anxiety among seafarers. This resonates with the feeling of seafarers as shared by R6:

*"The poor seaman you know after 4–5 month on board wants to come back home. And there were so many people stopped. I had experience I have some friend in another ship which they were in the China and their country does not allow them to come back to home for first month. ".*

*(R6)*

Repatriation is a right of people which is also mentioned in the MLC 2006 and states have agreed to follow it. However, the problem of repatriation as reported by seafarers and

managers of the company might be due to the port restriction. Respondents believed that it was not in the hands of the company but of port authorities and local law. Similar sentiments were also expressed by managers of the companies. Though they tried their best, they could not repatriate the seafarers due to port authorities and local regulations. Here is the response from R2 supporting the argument:

*"From my point of view, the seafarer is a free nationality easily they can go in one, joining and signing also monitoring of MLC implementation. Most of the things were not in hand of the management of company. Most of the things were managed by the port authority and local regulation."*

*(R2)*

Seafarers also suggested that port authorities, state parties, and organizations be considerate about the situation and facilitate the easy repatriation of seafarers and support seafarers in hard times like this if similar incidents happen again. Here is what R8 suggests:

*"I think, if all countries or any companies and government or any organization issuing the rules and regulation in future if it is same situation or any cases like COVID19 happen again; I think they [state parties and organizations] have to take care of seafarers more and at least they allow sign off"*

*(R8)*

Seafarers and managers also questioned the role of international organizations like the International Maritime Organization (IMO). They felt IMO should have anticipated this and been proactive in dealing with the situation to ease the port restrictions. Seafarers and managers felt IMO cared less about the health and wellbeing of seafarers when they needed IMO the most, particularly to facilitate repatriation during lockdown. Seafarers were not satisfied with support and facilitation to ease the situation for seafarers and R12 shares:

*"IMO does not perform anything regarding seafarers which I understand till now. IMO will make the sea life difficult than before and normally doesn't think about seafarers."*.

*(R12)*

Seafarers should have adequate opportunities to be in touch with their families and suggest organizations to provide facilities and opportunities to be in touch with their families and R9 requests:

*"If the IMO is trying to give some facilities to seafarers to be in touch with the family I don't know how. But if they are trying to give some link up communication to be in touch with the family at any time, little bit of pressure will be released."*.

*(R9)*

The findings of the present study demonstrated that the majority of seafarers were not satisfied with the guidelines to be followed, particularly at the beginning of the COVID-19 pandemic. Seafarers also reported that it was very hard to work with masks, restrictions, and social distancing. The port closure meant that they did not have access to the food of their choice, recreational activities, and medical facilities. Seafarers were counselled about the importance

of being safe and following guidelines. With time, they became more comfortable with the COVID-19 preventive measures and guidelines to be followed as R7 says:

*"Aah (takes long breathe) my experience probably during COVID-19 pandemic, it is so hard because every time you have to check everything. Around yourself, around your colleague, around your family and you have to follow the social distance and every time you have to take care regarding touching any area. This is too hard."*

*(R7)*

*3.2.2.3. Adaptation and case management.* Although the respondents did not report any on-board cases and their first-hand experience, there were some cases on board as reported by the manager. The seafarers also reported that after hearing about cases on other ships, they were more serious about following the guidelines and staying safe. The companies did prepare COVID-19 outbreak management plans and that helped them to deal with the pandemic. The companies already have an ERS, and they add components of COVID-19 to prepare a COVID-19 outbreak management plan. Regarding the management of the case on board, R11 says:

*"But before joining we had some cases for example the third parties or the repairman who were boarding, we had some case like this. When the crew check their temperature and they took the test. Its result was positive for one of the repairmen. The crew fortunately, we did not have anyone because it was close area, and no one was infected. ".*

*(R11)*

Benefit was also another aspect that was mentioned by participants which also highlights the effectiveness of the guidelines.

## 4. Discussion

This qualitative research has explored the implementation of COVID-19 guidelines and their effect on the seafarers' health and wellbeing. The development of guidance about COVID-19 started only six months after WHO declaration of COVID-19 as a pandemic. The guidance developed was revised only after a few months of issue to address the COVID-19 related issues and seafarers' needs are reflective of delayed response [10, 11]. Delayed response by the international organization and state parties to the needs and problems faced by seafarers has been linked to an increased risk of infection, death, and deterioration of the seafarer's mental health [4, 7, 8]. This also shows weak monitoring, evaluation, feedback and learning mechanisms among state parties and maritime organization. So far, all the guidance has been revised to fulfil the gap which the first version of guidance had. The frequent revision of guidance over the period of time showcases how the situation of COVID-19 has also been uncertain for state parties and international organizations to deal with and to learn from.

The majority of seafarers in this study were found to be dependent on the media to bridge the gap of knowledge and stated they were not sure about the authenticity and accuracy of information, which often increased their stress. The findings obtained from a cross-sectional study conducted on cargo seafarers also showed that the excessive use of social media was correlated with compromised mental health [20]. Consistent adherence to guidelines ensuring social distancing and uniform standard preventive practices can aid in the reduction of infection on board. In their study, Battineni et al. also found that shipping companies must come

up with suitable guidelines, programs and interventions to prevent the spread of COVID-19 which is in agreement with our study [21].

In this study, seafarers admitted their habituation to certain ways of living and working conditions and reported that their dissatisfaction with following the guidelines as the preventive measures for COVID-19 made their work-life more difficult. Similar dissatisfaction and experience were also observed among those working in metal refining with the use of PPE [22] while the insecurity of seafarers towards available safety mechanisms during the COVID-19 pandemic has also been reported by the seafarer's Happiness Index Report 2020 too [23]. On the other hand, this reflects the need of occupational health interventions and psychosocial counselling [21]. The unique finding of this study suggested that seafarers were also concerned and determined not to get infected and return to their home safely and the motivated participants of this study were to follow COVID-19 guidelines.

The results of the current study also demonstrated that the instructions, orientation, and training regarding the use of guidelines by the master may not adequately address the needs of seafarers during such a crisis. This study suggests behaviour-changing communication or effective infection control training during humanitarian crisis rather than general instructions and guidelines to help seafarers adapt easily to the situation. This finding of the current study is consistent with those of a rapid review conducted on healthcare workers suggesting a reduced risk of COVID-19 infection due to effective infection control training [24].

According to the findings of this study, restriction on the movement, recreational activity, and limited choice of food of seafarers are observed during lockdown. In addition, it was also evident that the participants who were at home during lockdown could not join the vessel and they were not able to earn money to support their families during such a crisis. Furthermore, uncertainties of when the seafarers could repatriate and use of substance abuse on board also pushed seafarers to an increased risk of accidents and injuries compromising the occupational health of seafarers. Sagaro GG et al. reports that injury and trauma were the second most assisted medical conditions through telemedicine to seafarers suggests compromised health and wellbeing of seafarers which is in agreement with our study the [25].

In this study, one of the major factors affecting the seafarers' mental health and wellbeing was access to services at the port and uncertain repatriation. Port restrictions, difficulties in repatriation, limited sea-shore interactions during the COVID-19 pandemic and their effects on health and wellbeing of seafarers have been investigated by previous studies [5, 26, 27]. The MLC 2006, International Health Regulations (IHR), Safety of Life at Sea (SOLAS) convention and Standards of Training, Certification and Watch keeping for Seafarers (STCW) convention have clearly stated that seafarers are entitled to medical care and assistance. However, participants were denied their access to medical services at port during the COVID-19 pandemic while some of the seafarers with injuries were even declined services at the port-by-port authorities. Thus, seafarers were deprived of their rights and entitlement to access medical services at port. In their study, Paukszat et al. showed the limited shore interaction for medical care while Tzivakaou highlighted seafarer being denied the right to seek emergency medical services [4, 28]. Effort from state parties to ease the accessibility of medical service varied port to port even though MLC 2006 identifies access to medical care, health protection and welfare measures and other social measures as rights of seafarers [29].

The impact of the implementation of the COVID-19 guidelines varied according to the experience of seafarers. In the present study, the majority of the participants reported that their stress and anxiety decreased after the implementation of the COVID-19 guideline, while others did not feel any difference. The possible reason for such an impression might be due to unmet needs of seafarers that the guidelines did not address (e.g., repatriation, access to port services, imposed restrictions and motivation). The assessment of needs and participatory

approach in the development of guidelines could improve their applicability and adaptability. COVID-19 could affect seafarers' mental health and wellbeing, but little attention was given to exploring it by the shipping companies. There were hotline numbers, access to psychologists and online platforms to promote the seafarers' mental health [10, 11, 29]. However, no emphasis on the assessment of the mental health of seafarers by companies or experience of seafarers using those services was observed in this study. In addition, the guidelines produced by international organizations and shipping companies have hardly consulted the seafarer to include need-based components for better health and wellbeing of seafarer. This study revealed that the focus of companies was to prevent the spread of COVID-19 or case management and less on the improved health and wellbeing of seafarers, which was consisted with the study of Nitin Mukesh [5]. This study is one of the few qualitative studies invigilating the effects of implementation of COVID-19 guidelines on health and wellbeing of seafarers. Furthermore, the results of this study can be an important reference to strengthen occupational health and management mechanisms in the Maritime sector and to respond to occupational health risk factors caused by pandemics and emergencies in future. The limitation of this study is that all respondents were male seafarers. So, this study did not explore female seafarers' experiences regarding COVID-19.

## 5. Conclusion

Seafarers' life has always been restrictive and isolated. Seafarers were stranded at sea for multiple months away from their family and friends. Seafarers were afraid of infection and new variants of COVID-19, stressed about the extra precautions and restrictions, stressed about the long working hours and delay in repatriation and worried about the family. The implementation of COVID-19 guidelines and protocols were introduced to facilitate the difficulties faced by seafarers and to enhance the health and wellbeing of seafarers but its impact on seafarers' health and wellbeing is unknown. Thus, this study has explored the experiences of seafarers implementing COVID-19 guidance.

The findings of this study concluded that implementation of the guidelines were effective in preventing the spread of COVID-19, reducing fear and promoting uniformity in practice when they were available. The guidelines helped to reduce anxiety and stress in seafarers as they followed the step-to-step prevention strategies provided by the guidance. However, seafarers reported that the guidance did not adequately address their mental health issues such as repatriation, fear of family, re-joining the ship, fear of COVID-19 infection and motivations such as bonuses, increased salary, and unlimited internet facilities. Lack of urgency, coordination, and early effort, as well as poor accountability, monitoring, and evaluation from state members and international organizations to rescue seafarers from the effects of pandemic and lockdown, has had a severe impact on seafarers' health and well-being. As a result, this study recommends: meaningful participation of seafarers in the development and revision of guidance; assessing seafarers' health needs; providing incentives; facilitating repatriation process and mainstream monitoring; evaluation and accountability mechanism of interventions; appropriate and adequate training for seafarers; and using the BCC approach during such pandemic. The findings of this study are noble and unique, which can help bridge the existing information gap regarding implementation of COVID-19 guidance and can help as an important reference to strengthen occupational health and management mechanisms at ship and to respond to occupational health risk factors in future emergencies.

## Author Contributions

**Conceptualization:** Fereshteh Baygi.

**Data curation:** Amit Timilsina, Fereshteh Baygi.

**Formal analysis:** Amit Timilsina, Fereshteh Baygi.

**Methodology:** Amit Timilsina.

**Project administration:** Amit Timilsina.

**Software:** Amit Timilsina.

**Supervision:** Fereshteh Baygi.

**Visualization:** Amit Timilsina, Fereshteh Baygi.

**Writing – original draft:** Amit Timilsina.

**Writing – review & editing:** Amit Timilsina, Fereshteh Baygi.

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
