## [Decision Letter · Decision Letter 0]

28 Aug 2022

PONE-D-22-16406COVID-19 guidelines and its effect on seafarers’ health and wellbeingPLOS ONE

Dear Dr. Timilsina,

Thank you for submitting your manuscript to PLOS ONE. After careful consideration, we feel that it has merit but does not fully meet PLOS ONE’s publication criteria as it currently stands. Therefore, we invite you to submit a revised version of the manuscript that addresses the points raised during the review process.

We look forward to receiving your revised manuscript.

Kind regards,

Tauseef Ahmad

Academic Editor

PLOS ONE

https://journals.plos.org/plosone/s/file?id=ba62/PLOSOne_formatting_sample_title_authors_affiliations.pdf".

“Authors declare no competing interests exist.”

4. Please upload a copy of Figure 1, to which you refer in your text on page 7. If the figure is no longer to be included as part of the submission please remove all reference to it within the text.

Additional Editor Comments:

A lot of improvements need to be made to the manuscript.

Reviewers' comments:

Reviewer's Responses to Questions

**Comments to the Author**

1. Is the manuscript technically sound, and do the data support the conclusions?

Reviewer #1: Partly

Reviewer #2: Yes

2. Has the statistical analysis been performed appropriately and rigorously? 

Reviewer #1: N/A

Reviewer #2: N/A

3. Have the authors made all data underlying the findings in their manuscript fully available?

Reviewer #1: Yes

Reviewer #2: Yes

4. Is the manuscript presented in an intelligible fashion and written in standard English?

Reviewer #1: No

Reviewer #2: Yes

5. Review Comments to the Author

Reviewer #1: Literature review is inadequate

Methodology

• Several unclear statements. Example is on page 4“Seafarers who were on board during or after these

response mechanisms were recruited”.

• The verbal informed consent was obtained from all participants before the interview and the objectives of the research were explained to them. This is generally not the case, explanation should come before consent is granted.

• Triangulation with only one method of data collection is misleading.

• The methodology is silent on the sampling techniques used. Another flaw is the general lack of justification for the choices made.

• The triangulation of data was done by interviewing three managers and reviewing the guidelines. I don’t see the review of the guidelines

• There is no figure 1 in the document as indicated on page 7.

Presentation

• Some of the point discussed were not mentioned by the participants in the study conducted. Example, participant being denied access to medical services.

Others

• No implication

• No proper recommendation

• Weak conclusion.

• The language is unsatisfactory. Proofreading is required.

Reviewer #2: First, I apologize for the delay in sending my peer review.

This study is a qualitative study of the impact of the COVID-19 guidelines on the health and well-being of seafarers, and given that the COVID-19 guidelines affect people in a variety of occupations, we assert that this qualitative study is important.

However, I believe that the paper would be even better if the results from this qualitative study were presented in a model.

Also, would it be difficult to present the details of each of the 13 interviewees in a table, rather than presenting them as a group, e.g., "age: 29-42"?

6. PLOS authors have the option to publish the peer review history of their article (what does this mean?). If published, this will include your full peer review and any attached files.

Reviewer #1: No

Reviewer #2: **Yes: **Junko Okuyama

---

## [Author Response · Author response to Decision Letter 0]

7 Nov 2022

31st October 2022

To,

The editor,

Plos One.

Subject: Regarding revision and submission of response letter (PONE-D-22-16406).

Dear editors,

We would like to re-submit our original article entitled COVID-19 guidelines and its effect on seafarers' health and well-being after addressing all the comments recommended and suggested by the reviewers. The comments and feedback from the reviewers has definitely been helpful to refine this article and we would like to take this opportunity to thank both of our reviewer for valuable insights and feedback. As stated by the reviewer, this article is very noble and important to bridge the existing gap regarding implementation of COVID-19 guidelines. Thus, we would like to request for the acceptance of this article for the publication. 

Also, we author declare no conflict of interest whatsoever and request to edit in the online portal as well. If the editorial team and reviewer team has any other feedback for us, we would be happy to receive and work on them.

Please find attached point to point comments' response (in red font color) attached below. 

Regards,

On behalf of co-authors

Amit Timilsina

timilsinaamit@gmaill.com

Point to point response of comments (PONE-D-22-16406)

Response: The style requirement as suggested by the editors has been addressed in the manuscript.

“Authors declare no competing interests exist.”

Response: The competing interest has been edited in the cover letter and as suggested by the editor, we request to change the online submission form on our behalf. 

Response: The ethics statement has been mentioned now only in method section. Previously, it was mentioned in the declaration section under ethics approval and consent to participant. It has been removed from this section. Only the statement regarding approval from University has been mentioned in the declaration section.

4. Please upload a copy of Figure 1, to which you refer in your text on page 7. If the figure is no longer to be included as part of the submission please remove all reference to it within the text.

Response: Figure 1 was already submitted as a separate file in the online submission form. It definitely needs to be included as part of submission. 

Additional Editor Comments:

Reviewer's Responses to Questions

5. Review Comments to the Author

Reviewer #1: Literature review is inadequate

Response: This study is nobel as there are very few research conducted on effective implementation of COVID-19 guidance's and protocols. The literatures available for this study topic are very few. Literature review for this topic was revisited and some of the new researches or updates were found and added accordingly to this paper. 

Methodology

• Several unclear statements. Example is on page 4“Seafarers who were on board during or after these

response mechanisms were recruited”.

Response: The methodology section has been reviewed again and appropriate changes have been made which is reflected in track change and clean version.

• The verbal informed consent was obtained from all participants before the interview and the objectives of the research were explained to them. This is generally not the case, explanation should come before consent is granted.

Response: The explanation and consent process was completed prior to interview date. However, consent was verbally retaken while starting the interview as well. The language has been reworked and made clearer. 

• Triangulation with only one method of data collection is misleading.

Response: Since, the topic of this study demands review of available guidances and study related to similar topic; systematic literature review was also conducted as part of triangulation. This section has been added to the methodology section acknowledging valuable suggestion by the reviewer. Thus, IDI and in-depth literature review, two different methods of triangulation suffice methodologically and is not misleading.

• The methodology is silent on the sampling techniques used. Another flaw is the general lack of justification for the choices made.

Response: The sampling technique 'Purposive sampling' and its process have now been clearly mentioned in the manuscript. 

• The triangulation of data was done by interviewing three managers and reviewing the guidelines. I don’t see the review of the guidelines

Response: The short section regarding review of guideline was explained in the beginning of discussion section previously. In response to the comment, we have not written a short separate section on findings of guidance review. It has also been now explained explicitly in methodology section too. 

• There is no figure 1 in the document as indicated on page 7.

Response: The figure was submitted separately in the submission form. 

Presentation

• Some of the point discussed were not mentioned by the participants in the study conducted. Example, participant being denied access to medical services.

Response: Not all quotes can be included within the manuscript given the word limitation for the manuscript. However, the important excerpts for denial of medical services has been now included in the manuscript.

Others

• No implication

• No proper recommendation

• Weak conclusion.

• The language is unsatisfactory. Proofreading is required.

Response: The recommendation has been updated within the discussion section, conclusion has been revised and re-written, implication of this study has been well explained at the end of discussion section and in the conclusion section. Similarly, proof-reading has been made by us and also reviewed by a professional. 

Reviewer #2: First, I apologize for the delay in sending my peer review.

This study is a qualitative study of the impact of the COVID-19 guidelines on the health and well-being of seafarers, and given that the COVID-19 guidelines affect people in a variety of occupations, we assert that this qualitative study is important.

However, I believe that the paper would be even better if the results from this qualitative study were presented in a model.

Also, would it be difficult to present the details of each of the 13 interviewees in a table, rather than presenting them as a group, e.g., "age: 29-42"?

Response: 

The figure 1 illustrates the model of the study which is based on one generic research statement, two themes and six subthemes. 

Having detailed information for each interviewees may lead to compromised anonymity of the participants, thus we chose to keep it as a range rather than individual socio-demographic information. For example, a seafarer of age 32 working at desk of 8 years of experience might help other know the participant. Thus, we want to anoynymize participants' information. 

6. PLOS authors have the option to publish the peer review history of their article (what does this mean?). If published, this will include your full peer review and any attached files.

Response: Yes, our identity can be revealed.

---

## [Decision Letter · Decision Letter 1]

15 Dec 2022

PONE-D-22-16406R1COVID-19 guidelines and its effect on seafarers’ health and wellbeingPLOS ONE

Dear Dr. Timilsina,

Thank you for submitting your manuscript to PLOS ONE. After careful consideration, we feel that it has merit but does not fully meet PLOS ONE’s publication criteria as it currently stands. Therefore, we invite you to submit a revised version of the manuscript that addresses the points raised during the review process.

We look forward to receiving your revised manuscript.

Kind regards,

Tauseef Ahmad

Academic Editor

PLOS ONE

Journal Requirements:

Additional Editor Comments (if provided):

The manuscript has been improved. There are still some minor revisions that one of the reviewers needs to request before further processing can proceed.

Reviewers' comments:

Reviewer's Responses to Questions

**Comments to the Author**

1. If the authors have adequately addressed your comments raised in a previous round of review and you feel that this manuscript is now acceptable for publication, you may indicate that here to bypass the “Comments to the Author” section, enter your conflict of interest statement in the “Confidential to Editor” section, and submit your "Accept" recommendation.

Reviewer #1: All comments have been addressed

Reviewer #2: All comments have been addressed

2. Is the manuscript technically sound, and do the data support the conclusions?

Reviewer #1: Yes

Reviewer #2: Yes

3. Has the statistical analysis been performed appropriately and rigorously? 

Reviewer #1: N/A

Reviewer #2: Yes

4. Have the authors made all data underlying the findings in their manuscript fully available?

Reviewer #1: Yes

Reviewer #2: Yes

5. Is the manuscript presented in an intelligible fashion and written in standard English?

Reviewer #1: Yes

Reviewer #2: Yes

6. Review Comments to the Author

Reviewer #1: The manuscript has been improved. The weakness in the methodology is addressed. However, I believe a bit more literature on Covid versus stress could be added. This literature does not necessary have to be on seafarer's.

Reviewer #2: The authors responded sincerely to the reviewers' comments and improved the manuscript.

As the authors note, seafarers are under a great deal of stress, being away from their families for long periods of time.

In such a work situation, I belieI look forward to the authors' future research developments.

I look forward to the authors' future research developments.

7. PLOS authors have the option to publish the peer review history of their article (what does this mean?). If published, this will include your full peer review and any attached files.

Reviewer #1: No

Reviewer #2: **Yes: **Junko Okuyama

---

## [Author Response · Author response to Decision Letter 1]

23 Jan 2023

15th January 2023

To,

The editor,

Plos One.

Subject: Regarding revision and submission of response letter (PONE-D-22-16406).

Dear editors,

We would like to re-submit our original article entitled COVID-19 guidelines and its perceived effect on seafarers' health and well-being: a qualitative study after addressing all the comments recommended and suggested by the reviewers/editors. There was one comment from the editorial team which is to revise the reference section. This has been updated and revised. The reviewer has mentioned that all the comments and changes have been made. Similarly, as stated by the reviewer, this article is very noble and important to bridge the existing gap regarding implementation of COVID-19 guidelines. Thus, we would like to request for the acceptance of this article for the publication. 

Also, we author declare no conflict of interest whatsoever and request to edit in the online portal as well. If the editorial team and reviewer team has any other feedback for us, we would be happy to receive and work on them.

The changes have been submitted in track change and clean version in the manuscript. Please find attached point to point comment response (in red font color) attached below. 

Regards,

On behalf of co-authors

Amit Timilsina

timilsinaamit@gmaill.com

Point to point response of comment (PONE-D-22-16406)

Response: The reference section has been revisited one by one, checked, revised wherever necessary and updated. One reference was removed and new reference was added while one reference's status was revised and updated.

---

## [Decision Letter · Decision Letter 2]

27 Mar 2023

COVID-19 guidelines and its perceived effect on seafarers’ health and wellbeing: a qualitative study

PONE-D-22-16406R2

Dear Dr. Timilsina,

We’re pleased to inform you that your manuscript has been judged scientifically suitable for publication and will be formally accepted for publication once it meets all outstanding technical requirements.

Kind regards,

Tauseef Ahmad

Academic Editor

PLOS ONE

Additional Editor Comments (optional):

There are several points that I believe have been adequately addressed by the authors in the revision. This highlights the concerns that I raised in the prior review.

Reviewers' comments:

Reviewer's Responses to Questions

**Comments to the Author**

1. If the authors have adequately addressed your comments raised in a previous round of review and you feel that this manuscript is now acceptable for publication, you may indicate that here to bypass the “Comments to the Author” section, enter your conflict of interest statement in the “Confidential to Editor” section, and submit your "Accept" recommendation.

Reviewer #2: All comments have been addressed

2. Is the manuscript technically sound, and do the data support the conclusions?

Reviewer #2: Yes

3. Has the statistical analysis been performed appropriately and rigorously? 

Reviewer #2: Yes

4. Have the authors made all data underlying the findings in their manuscript fully available?

Reviewer #2: Yes

5. Is the manuscript presented in an intelligible fashion and written in standard English?

Reviewer #2: Yes

6. Review Comments to the Author

Reviewer #2: The main findings of this paper are important for maintaining health and wellbeing during the COVID-19 pandemic in an environment where seafarers are not free to leave their workplace.

7. PLOS authors have the option to publish the peer review history of their article (what does this mean?). If published, this will include your full peer review and any attached files.

Reviewer #2: **Yes: **Junko Okuyama

---

## [Editor Report · Acceptance letter]

31 Mar 2023

PONE-D-22-16406R2 

COVID-19 guidelines and its perceived effect on seafarers’ health and wellbeing: a qualitative study 

Dear Dr. Timilsina:

I'm pleased to inform you that your manuscript has been deemed suitable for publication in PLOS ONE. Congratulations! Your manuscript is now with our production department. 

Kind regards, 

on behalf of

Dr. Tauseef Ahmad 

Academic Editor

PLOS ONE